# Validity of the International Fitness Scale (IFIS) and its associations with cardiometabolic health and body composition in adults with type 2 diabetes: A cross-sectional study

Ángel Herraiz-Adillo[1], Karin Rådholm[2,3], Fredrik Iredahl[4,5], Carl-Johan Carlhäll[1,6], Nahuel Roemkens[1], Mikael Forsgren[1,6,7], Olof Dahlqvist Leinhard[1,6,8], Peter Lundberg[1,6,8,9], Stergios Kechagias[1], Nils Dahlström[1,6,8], Patrik Nasr[1,5,6], Mattias Ekstedt[1,6*☯], Pontus Henriksson[1*☯]

**1** Department of Health, Medicine and Caring Sciences, Linköping University, Linköping, Sweden, **2** Primary Health Care Center Kärna, and Department of Health, Medicine and Caring Sciences, Linköping University, Linköping, Sweden, **3** The George Institute for Global Health, University of New South Wales, Sydney, Australia, **4** Primary Health Care Center Åby, and Department of Health, Medicine and Caring Sciences, Linköping University, Linköping, Sweden, **5** Wallenberg Center for Molecular Medicine (WCMM), Linköping University, Linköping, Sweden, **6** Center for Medical Image Science and Visualization (CMIV), Linköping University, Linköping, Sweden, **7** AMRA Medical AB, Linköping, Sweden, **8** Clinical Department of Radiology in Linköping, Region Östergötland, Linköping, Sweden, **9** Clinical Department of Medical Radiation Physics, Region Östergötland, Linköping, Sweden

☯ These authors contributed equally to this work and are shared senior authors on this work.
* pontus.henriksson@liu.se (PH); mattias.ekstedt@liu.se (ME)

## Abstract

### Aims

To assess the validity of the International Fitness Scale (IFIS) for evaluating cardiorespiratory fitness compared to the 6-minute walk test (6MWT) (criterion validity) and to examine associations with cardiometabolic and body composition outcomes (construct validity).

### Methods

A cross-sectional analysis was conducted on 282 adults with type 2 diabetes mellitus (T2DM) (mean age 63.6±8.1 years, 37.9% women). Self-reported fitness was assessed using IFIS, including overall, cardiorespiratory, muscular, speed-agility and flexibility scores. Objective cardiorespiratory fitness was assessed using the 6MWT. Associations with 6MWT, cardiometabolic (i.e., cardiovascular health score, metabolic dysfunction-associated steatotic liver disease [MASLD], high-sensitivity C-reactive protein) and body composition outcomes (i.e., body mass index [BMI], visceral adipose tissue volume, thigh fat-free muscle volume) were analyzed using ANCOVA (adjusted by sex and age) and ROC curves, compared using DeLong tests.

**Data availability statement:** The data underlying this article cannot be made publicly available due to legal restrictions and the need to protect participant privacy. However, information on how to request access to the data in accordance with the guidelines of the Swedish Ethical Review Authority (email: registrator@etikprovning.se) can be obtained by contacting the study organization, Region Östergötland (email: region@regionostergotland.se).

**Funding:** The EPSOMIP study was supported by ALF Grants, Region Östergötland (ME, FI, PL, PN), unrestricted grants from GILEAD (ME) and Diapharma (SK), the Lion Research Grant from the Faculty of Medicine, Linköping University (PN), the Swedish Research Council (VR 2020-04826 to PL) and the Swedish Medical Association (SLS-960199 to KR). This specific study was additionally supported by ALF Grants, Region Östergötland (RÖ-999299 to PH). The funders had no role in study design, data collection and analysis, decision to publish, or preparation of the manuscript. There was no additional external funding received for this study.

**Competing interests:** MF and ODL are employees and shareholders of AMRA Medical AB. ME has received lecture fees from Novo Nordisk. PL is a minority owner of AMRA Medical AB. The EPSOMIP study was partly supported by unrestricted grants from GILEAD (ME) and Diapharma (SK). This does not alter our adherence to PLOS ONE policies on sharing data and materials. There are no patents, products in development or marketed products associated with this research to declare.

## Results

Both IFIS overall and IFIS cardiorespiratory showed graded associations with 6MWT ($p < 0.001$ for ANCOVA models). For discriminating performance below the population-predicted mean 6MWT distance, IFIS cardiorespiratory showed an AUC of 0.615 (95% CI: 0.552–0.678). Apart from IFIS muscular, IFIS scores were significantly associated with all cardiometabolic and body composition outcomes, with ROC analyses showing at least similar predictive performance for IFIS overall and cardiorespiratory fitness scores compared to 6MWT. Notably, IFIS overall outperformed the 6MWT for predicting poor cardiovascular health (AUCs: 0.678 [95% CI: 0.618–0.737] versus 0.586 [95% CI: 0.517–0.654]) and MASLD (AUCs: 0.630 [95% CI: 0.563–0.696] versus 0.519 [95% CI: 0.443–0.595]).

## Conclusion

This study supports that IFIS may be a practical tool for adults with T2DM, showing evidence of criterion validity via graded association with 6MWT performance, and construct validity via significant associations with cardiometabolic and body composition outcomes. Although its ability to discriminate impaired functional capacity was limited, IFIS may serve as a feasible alternative for fitness assessment when objective testing is not possible in primary care.

## Introduction

Type 2 diabetes mellitus (T2DM) is a growing global health concern, affecting more than 500 million people [1]. Its prevalence is rising alarmingly, with the disease burden, measured in DALYs, having tripled over the past 30 years [2]. While the etiology of T2DM is multifactorial, it is closely associated with lifestyle factors including obesity, poor diet, and physical inactivity. Individuals with T2DM are at a significantly higher risk of developing cardiovascular risk factors (e.g., hypertension, dyslipidemia, insulin resistance, and liver steatosis) and cardiovascular disease (CVD) [3].

Adequate physical fitness, particularly cardiorespiratory fitness, not only reduces the risk of developing T2DM [4,5] but also improves glycemic control, insulin sensitivity, and cardiovascular outcomes in those already diagnosed [6]. The American Heart Association recognizes cardiorespiratory fitness as a vital clinical sign and advocates for its routine assessment in clinical settings [7]. However, laboratory cardiopulmonary exercise testing, the gold standard for evaluating cardiorespiratory fitness, is often impractical in routine clinical practice due to substantial resource demands related to personnel, equipment, and costs. While some field exercise tests, including the 20-meter and the 6-minute walk tests [8], can objectively assess cardiorespiratory fitness, they may be impractical in clinical practice, particularly for patients with T2DM. These patients may experience conditions like peripheral neuropathy or peripheral arterial disease, which make walking tests challenging. This highlights

the need for more accessible and less resource-intensive methods, such as self-reported cardiorespiratory fitness assessments.

Previous studies have demonstrated that self-reported physical fitness, particularly through the use of the International Fitness Scale (IFIS), provides a valid approximation of objectively measured fitness levels. The IFIS has shown acceptable validity with objective fitness measures across diverse populations, including children, adolescents, and young adults [9,10]. Moreover, self-reported fitness assessed using the IFIS has shown associations with CVD risk factors that are comparable in both magnitude and direction to those observed with objective fitness measures [9–11]. Despite these encouraging findings, most studies employing the IFIS have been conducted in populations with relatively low CVD risk, such as younger individuals, leaving a gap in evidence for high-risk populations, particularly those with T2DM. Given the elevated CVD risk inherent in T2DM [12], it is essential to investigate whether the IFIS can reliably evaluate fitness and its relationship with CVD risk factors in this population. Furthermore, no previous study has directly compared the predictive ability of the IFIS versus objectively measured fitness for CVD risk, including cardiometabolic and body composition outcomes, in any population.

We hypothesized that the IFIS is a valid tool for assessing physical fitness in adults at high cardiovascular risk, particularly those with T2DM. Thus, the aims of this study were: 1) to investigate the validity of the IFIS self-reported physical fitness scores (i.e., overall and cardiorespiratory fitness) in assessing cardiorespiratory fitness considering the objectively measured 6-minute walk test as reference (criterion validity), and 2) to examine and compare the associations of the IFIS self-reported physical fitness scores and the 6-minute walk test with a wide range of cardiometabolic and body composition outcomes in adults with T2DM (construct validity).

In this study, particular emphasis was placed on the cardiorespiratory fitness component of the IFIS, as it reflects a construct similar to that assessed by the 6-minute walk test and is the component more strongly associated with CVD and mortality [13].

## Materials and methods

### Study design and sample

This study used cross-sectional data from the Evaluating the Prevalence and Severity Of MASLD (metabolic dysfunction-associated steatotic liver disease) In Primary care (EPSOMIP) study [14]. The original study protocol was published under the acronym "EPSONIP"; in the present work we use "EPSOMIP" to reflect updated MASLD terminology. The EPSOMIP study aimed to comprehensively explore the association between T2DM and metabolism-related conditions (i.e., MASLD, epicardial fat and CVD) using non-invasive advanced techniques. The present study focuses on an exploratory post hoc study within the EPSOMIP protocol.

Recruitment took place in Östergötland County, Sweden, primarily through four primary healthcare centers in the cities of Norrköping and Linköping. Individuals with T2DM were invited to participate during their routine annual check-ups with their diabetes nurse or treating physician. Due to slower enrolment during the coronavirus disease 2019 pandemic, additional invitations were sent by letter to patients registered at participating primary healthcare centers. The recruitment period spanned from 03/04/2019–22/09/2023.

In the EPSOMIP protocol, inclusion criteria included a diagnosis of T2DM based on current guidelines [15] and an age range of 35–75 years. Exclusion criteria encompassed previously diagnosed liver cirrhosis or other primary liver diseases aside from MASLD, contraindication for performing Magnetic Resonance Imaging (MRI) (i.e., pacemaker, ferrous metal implants/fragments, claustrophobia, extreme obesity, and/or pregnancy), alcohol dependence, and inability to obtain Proton Density Fat Fraction (PDFF) measurements. Of the recruited participants, 18 withdrew consent, leaving 308 individuals enrolled in the EPSOMIP study. For the present analysis, after applying additional exclusion criteria, we included 282 participants with complete data on exposures, covariates, and at least one cardiometabolic or body composition outcome of interest. A flowchart of the study design is presented in Fig 1.

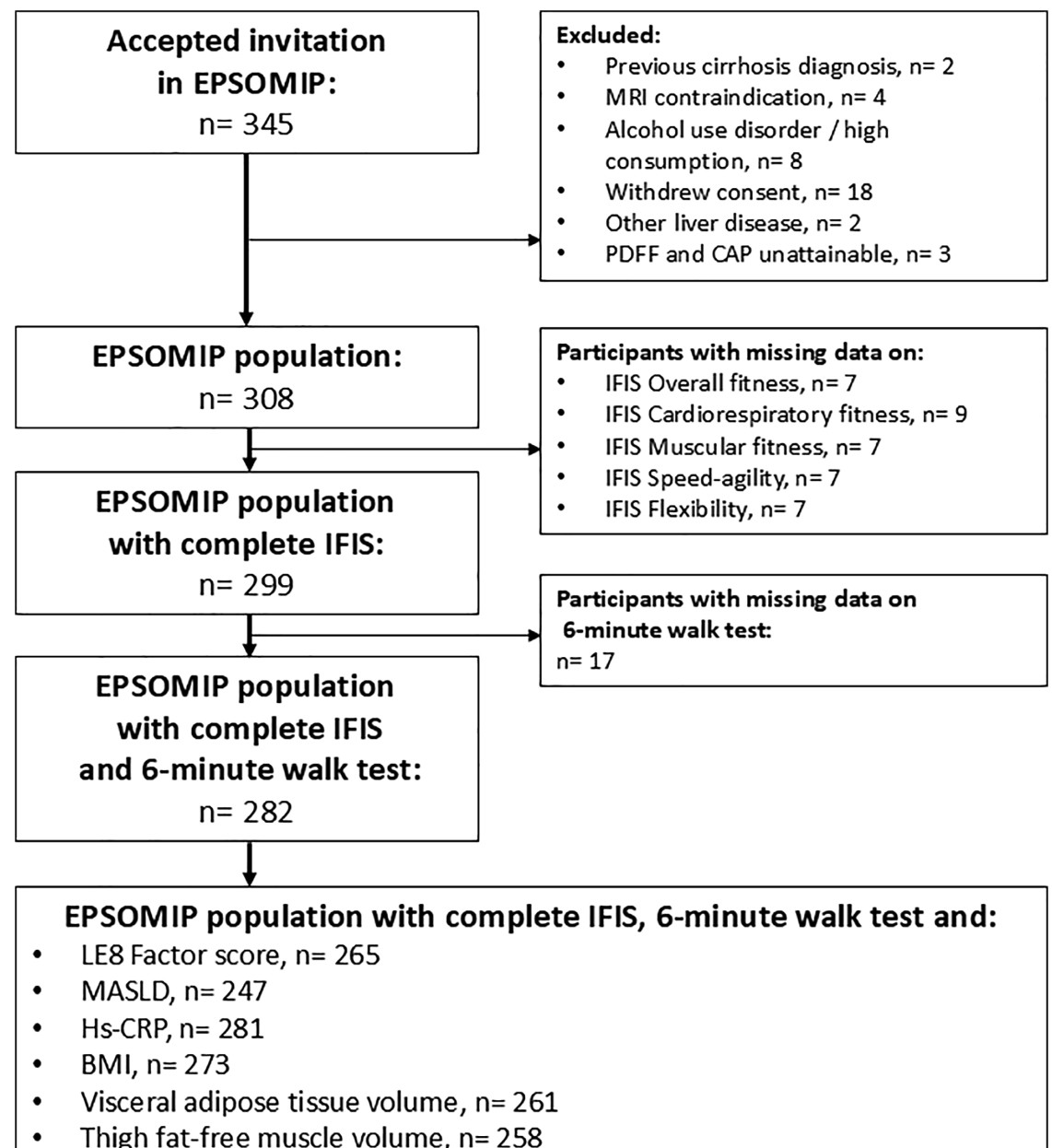

**Fig 1. Flow chart of the study.** BMI: body mass index, CAP: Controlled Attenuation Parameter, EPSOMIP: Evaluating the Prevalence and Severity Of MASLD in Primary Care, hs-CRP: high-sensitivity C-reactive protein, IFIS: International Fitness Scale, LE8: Life's Essential 8, MASLD: metabolic dysfunction-associated steatotic liver disease, MRI: Magnetic Resonance Imaging, PDFF: Proton Density Fat Fraction.

The Regional Ethical Board of Östergötland, Sweden, granted ethical approval for the EPSOMIP study (2018/176–31 and 2018/494–32), which also was prospectively registered (ClinicalTrials.gov identifier: NCT03864510). Written informed consent was obtained from each participant prior to inclusion, in accordance with the Declaration of Helsinki and Swedish accepted practices. Data collection and handling adhered to the General Data Protection Regulation (EU) 2016/679 and followed the standards of Good Clinical Practice (ICH-GCP).

## Physical fitness

**Self-reported physical fitness (IFIS).** Self-reported fitness was evaluated using the IFIS questionnaire, a brief, self-administered questionnaire originally validated in adolescents [10]. It takes approximately 1–2 minutes to complete and does not require trained personnel. The IFIS consists of five single-item questions, each addressing a different domain of perceived physical fitness: overall fitness, cardiorespiratory fitness, muscular fitness, speed-agility, and flexibility. Participants rate their fitness in each domain compared to peers of the same age, using a 5-point Likert scale: very poor (1), poor (2), average (3), good (4), and very good (5). This results in five separate scores, each ranging from 1 to 5, where higher scores indicate better perceived fitness in the respective domain. No established clinical cutoffs exist for the IFIS. For this study, the Swedish version of the IFIS questionnaire was used (available at: https://profith.ugr.es/ifis) with minor modifications to adapt the language for an adult population. The Swedish version has been validated in women during early pregnancy [11], but not, to our knowledge, in adults with T2DM, the specific gap this study explores.

**Objectively measured cardiorespiratory fitness (6-minute walk test).** Cardiorespiratory fitness was objectively evaluated using the 6-minute walk test, a submaximal exercise test that measures the distance walked in meters over six minutes [10,16], with proven validity across different populations [17,18]. The test was conducted in a corridor with a flat, hard surface at the participants' study centers as part of the EPSOMIP study. Participants were instructed to walk at a self-selected pace, allowing them to adjust their intensity and take breaks if needed. Standardized phrases of encouragement were provided at regular intervals to maintain motivation. The 6-minute walk test was used as a continuous variable and was further categorized as: < 25th percentile (197–488 meters), 25th-75th percentile (489–595 meters) and >75th percentile (596–794 meters).

## Cardiometabolic outcomes

The cardiovascular health score (Life's Essential 8), liver fat, and high-sensitivity C-reactive protein were considered relevant markers for assessing cardiometabolic outcomes. The cardiovascular health score effectively predicts CVD and mortality [19,20], while liver fat is linked to CVD and T2DM risk [21], and high-sensitivity C-reactive protein reflects inflammation, a key driver of CVD conditions [22].

Cardiovascular health was assessed using the Life's Essential 8 score, as defined by the American Heart Association [23]. In the present study, we focused specifically on Life's Essential 8 factors (BMI, blood lipids, blood pressure, and diabetes) rather than Life's Essential 8 behaviors (diet, sleep, physical activity, and smoking), as factors more directly reflect cardiometabolic outcomes and are more likely to be influenced by physical fitness. The Life´s Essential 8 Factor score accounts for actual values of non-HDL cholesterol, blood pressure, and blood glucose, along with the use of medication for these factors, as well as values of BMI. A detailed description of the scoring methodology has been provided as Supporting Information (Appendix 1). Briefly, and in line with the American Heart Association recommendations, each component was scored from 0 (poorest cardiovascular health) to 100 (best cardiovascular health). The Life's Essential 8 Factor score was then calculated as the unweighted average of the four components, yielding an overall score ranging from 0 to 100. Additionally, a normalized Life's Essential 8 Factor score was calculated by summing the Z-scores of the four components, which was then standardized into a single Z-score.

Liver fat (%) was quantified based on the hepatic triglyceride concentration, measured using PDFF by Magnetic Resonance Spectroscopy (MRS) [14,24]. Data were acquired using a 1.5T Achieva dStream MR scanner (Philips Healthcare, Best, The Netherlands).

High-sensitivity C-reactive protein was measured in blood serum samples collected after an overnight fast. The lower detection limit for high-sensitivity C-reactive protein was 0.15 mg/L; values below this threshold (n = 7) were assigned a value of 0.10 mg/L.

## Body composition outcomes

Body composition markers like BMI, visceral fat, and muscle volume are strongly linked to T2DM and cardiovascular risk. BMI indicates obesity, visceral fat is closely associated to metabolic dysfunction [25], and low muscle mass is tied to impaired glucose metabolism and poorer health outcomes [26,27].

BMI was measured using standardized procedures, calculated as weight divided by height squared, and expressed in units of kg/m². Visceral adipose tissue volume (measured in liters, and henceforth referred as visceral adipose volume) and thigh fat-free muscle volume (measured in liters, and henceforth referred as muscle volume) were analyzed using AMRA® Researcher (AMRA Medical AB, Sweden) based on whole-body water-fat separated dual-echo Dixon MR images acquired using a 1.5T Achieva dStream MR scanner (Philips Healthcare, Best, The Netherlands) [28]. Visceral adipose volume was considered as a continuous variable, while for each participant in EPSOMIP, a sex, height, and BMI invariant muscle volume Z-score was also calculated [29]. The muscle volume Z-score indicates how many standard deviations (SDs) the muscle volume of a participant deviates from what is expected given their sex and body size by comparing each participant to a matched group from the general population.

## Statistical analysis

We examined the distribution of continuous outcomes using both statistical (Kolmogorov–Smirnov test) and graphical (normal probability plots) methods. Variables that did not follow a normal distribution (i.e., high-sensitivity C-reactive protein and liver fat) were natural-logarithmically transformed prior to analysis.

The validity of the IFIS scores in classifying T2DM participants into appropriate cardiorespiratory fitness levels was assessed using analysis of covariance (ANCOVA), both with and without adjustments for sex (categorical) and age (continuous). The 6-minute walk test served as the dependent variable, while the IFIS test categories were treated as fixed factors. Additionally, ANCOVA models adjusted for sex and age were used to investigate the associations between both self-reported and objectively measured fitness tests with cardiometabolic and body composition outcomes. For ANCOVA, all six outcomes were standardized to study-specific Z-scores to facilitate comparison between them (including standardization of the Life's Essential 8 Factor Z-score and muscle volume Z-score).

To assess the diagnostic performance of the IFIS cardiorespiratory fitness item, we evaluated sensitivity, specificity, likelihood ratios, diagnostic odds ratios, and predictive values. Two thresholds were used to define reduced 6-minute walk test performance:

i) a population-based threshold, defined as performance below the population-predicted mean 6-minute walk test distance (522.9 meters), calculated using the Enright & Sherrill equations based on the weighted average of age, height, and weight for men and women in the study population; and ii) an individualized threshold, also based on the Enright & Sherrill equations, which account for each participant's age, sex, height, and weight. Functional impairment was defined as walking less than 80% of the individualized predicted distance [30]. Areas under the receiver operating characteristic (ROC) curves (AUCs) were calculated based on these thresholds to provide complementary insights into functional capacity. The predictive ability of self-reported and objectively measured fitness tests for cardiometabolic and body composition outcomes was evaluated by comparing the AUCs of unadjusted ROC curves using DeLong tests. Additionally, predicted probabilities for health outcomes were generated using logistic regression models that incorporated self-reported or objective tests as predictors, adjusted for age and sex. The AUCs of those predicted probabilities were also compared using DeLong tests. For generating ROC curves, we selected clinically relevant outcomes as follows: poor cardiovascular health (<50 points in consonance with the recommendation of the American Heart Association) [23], MASLD (≥5% of liver fat) [31], high-risk high-sensitivity C-reactive protein (>3.0 mg/L) [32], obesity (BMI ≥ 30 kg/m²), high visceral adipose volume (≥75th percentile), and low muscle volume <25th percentile in a general population (<−0.68 SDs) [33].

To verify the robustness of our results, a series of sensitivity analyses were conducted: i) excluding presumably low-intensity 6-minute walk tests defined as those with heart rate after the test below 60% of the predicted maximal heart

rate (using Tanaka's formula; predicted maximal heart rate = 208 − 0.7 × age), as those may indicate insufficient effort and affect test validity (n = 79) [34], and ii) considering different cut-offs to dichotomize the outcomes in ROC curves, specifically all Z-scores for the outcomes were dichotomized at the median (0 indicating worse health, 1 indicating better health).

To explore the contribution of T2DM-related conditions, i.e., neuropathy and angiopathy, to the criterion validity of IFIS, a complementary analysis was conducted using adjusted ANCOVA models comparing the 6-minute walk test performance across IFIS overall and IFIS cardiorespiratory fitness categories between T2DM participants with and without neuropathy/angiopathy.

Analyses were restricted to participants with complete data on IFIS and the 6-minute walk test. For cardiometabolic and body composition outcomes, an available-case approach was used, including all participants with data available for each outcome.

All statistical tests were two-sided and p < 0.05 was considered statistically significant. Analyses were conducted using Stata V.18 (StataCorp 2021).

## Results

### Participants' characteristics

The characteristics of participants are presented in Table 1 (stratified by three IFIS overall fitness categories) and S1 Table (stratified by five categories).

Finally, 282 middle-aged participants with T2DM (37.9% women, mean age 63.6 ± 8.1 years) were included. Duration of T2DM was 8.4 ± 6.9 years, with 20.3% of participants receiving insulin and 87.5% receiving non-insulin treatments for T2DM.

The mean distance in the 6-minute walk test was 538.9 ± 86.2 meters. For clinical outcomes, the mean cardiovascular health score, BMI, and visceral adipose volume were 51.9 points, 29.4 kg/m², and 6.1 liters, respectively, while the median liver fat content and high-sensitivity C-reactive protein levels were 7.4% and 1.2 mg/L, respectively.

S1 Fig illustrates the distribution of responses across all five IFIS scores. Overall, the distributions were relatively symmetric across the scores. However, in the IFIS cardiorespiratory test, participants more frequently scored "very poor" (14.9%) and less frequently scored "very good" (1.4%) compared to the other scores.

### Self-reported fitness versus objective cardiorespiratory fitness

Fig 2 illustrates the curvilinear dose-response associations between the IFIS overall fitness and cardiorespiratory fitness scores and the 6-minute walk test.

Participants reporting better overall fitness generally achieved longer distances in the 6-minute walk test (p < 0.001 for ANCOVA; see S2 Table). Specifically, participants reporting "very poor", "poor", "average", "good" and "very good" on the IFIS overall fitness covered unadjusted mean distances of 508.2, 500.0, 533.1, 552.3 and 598.9 meters, respectively. A linear dose-response association was observed for IFIS cardiorespiratory fitness, with participants achieving unadjusted mean distances of 494.9, 531.5, 546.9, 579.6, and 624.5 meters, respectively, (p < 0.001 for ANCOVA). A sensitivity analysis excluding those participants with a heart rate after 6-minute test below 60% of the predicted maximal heart rate (n = 79) slightly attenuated the associations but did not alter the conclusions (S2 Fig).

To identify individuals with reduced 6-minute walk test performance, defined as either walking below the population-predicted mean on the 6-minute walk test or achieving less than 80% of their predicted distance, reporting "very poor", "poor" or "average" cardiorespiratory fitness on the IFIS showed a sensitivity of 93.9% (95% CI: 87.8, 97.5) and specificity of 19.0% (95% CI: 13.4, 25.8) for the former, and a sensitivity of 87.5% (95% CI: 61.7, 98.4) and specificity of 13.3% (95% CI: 9.4, 18.1) for the latter. The corresponding AUCs were 0.615 (95% CI: 0.552, 0.678) and 0.556 (95% CI: 0.404–0.708), respectively (see S3 Table).

**Table 1. Clinical characteristics of the participants by IFIS overall fitness scores, 3 categories.**

| | | All (n=282) | IFIS overall fitness (n=282) | | | |
|---|---|---|---|---|---|---|
| | | | Very poor/poor n=37 | Average n=132 | Good/very good n=113 | P value |
| **Sex (female, %)** | 282 | 107 (37.9) | 18 (48.6) | 51 (38.6) | 38 (33.6) | 0.256 |
| **Age (years)** | 282 | 63.6±8.1 | 60.1±10.0 | 63.4±8.0 | 65.1±7.1 | 0.004 |
| **Duration of T2DM (years)** | 279 | 8.4±6.9 | 7.3±4.6 | 8.4±7.6 | 8.9±6.8 | 0.468 |
| **Current use of diabetes medications** | | | | | | |
| Insulin | 271 | 55 (20.3) | 5 (14.3) | 32 (25.2) | 18 (16.5) | 0.277 |
| Metformin | 280 | 217 (77.5) | 27 (75.0) | 99 (75.6) | 91 (80.5) | 0.883 |
| Sulfonylurea | 274 | 25 (9.1) | 6 (16.7) | 12 (9.3) | 7 (6.4) | 0.208 |
| SGLT2-inhibitor | 272 | 80 (29.4) | 12 (32.4) | 42 (33.3) | 26 (23.9) | 0.160 |
| Incretin-based | 273 | 23 (8.4) | 4 (11.1) | 9 (7.0) | 10 (9.2) | 0.691 |
| **Smoking** | 282 | | | | | 0.130 |
| Never smoked | | 138 (48.9) | 14 (37.8) | 59 (44.7) | 65 (57.5) | |
| Current smoker | | 11 (3.9) | 2 (5.4) | 4 (3.0) | 5 (4.4) | |
| Ex-smoker | | 133 (47.2) | 21 (56.8) | 69 (52.3) | 43 (38.1) | |
| **Physical activity** | 240 | | | | | |
| Moderate-vigorous PA (time spent in 1–5 min bouts, min/day) | | 11.3±11.0 | 7.8±9.2 | 11.4±12.1 | 12.3±9.9 | 0.140 |
| Light PA (time spent in 10 min bouts, min/day) | | 3.9±6.5 | 3.6±8.4 | 2.8±4.1 | 5.4±7.8 | 0.014 |
| Sedentary time (time spent in 30 min bouts, min/day) | | 553.8±167.5 | 599.0±196.0 | 556.8±162.4 | 535.5±162.3 | 0.180 |
| **Diabetic foot risk** | 266 | | | | | 0.835 |
| Healthy | | 193 (72.6) | 22 (66.7) | 87 (71.9) | 84 (75.0) | |
| Neuropathy/angiopathy | | 54 (20.3) | 9 (27.3) | 25 (20.7) | 20 (17.9) | |
| Previous foot ulcers | | 19 (7.1) | 2 (6.1) | 9 (7.4) | 8 (7.1) | |
| **6-minute walk test (meters)** | 282 | 538.9±86.2 | 501.1±79.0 | 533.1±87.8 | 558.0±81.9 | 0.001 |
| **6-minute walk test** | 282 | | | | | 0.091 |
| <25th percentile; <489 meters | | 71 (25.2) | 14 (37.8) | 36 (27.3) | 21 (18.6) | |
| 25-75th percentile; 489–595 meters | | 142 (50.4) | 18 (48.6) | 66 (50.0) | 58 (51.3) | |
| >75th percentile; ≥596 meters | | 69 (24.5) | 5 (13.5) | 30 (22.7) | 34 (30.1) | |
| **IFIS overall fitness** | 282 | 3.3±0.8 | 1.9±0.3 | 3.0±0.0 | 4.1±0.3 | <0.001 |
| **IFIS cardiorespiratory fitness** | 282 | 2.5±0.9 | 1.4±0.5 | 2.2±0.7 | 3.2±0.8 | <0.001 |
| **IFIS muscular fitness** | 282 | 3.3±0.8 | 2.5±0.9 | 3.2±0.7 | 3.7±0.6 | <0.001 |
| **IFIS speed-agility** | 282 | 2.9±0.9 | 2.0±0.8 | 2.7±0.8 | 3.4±0.7 | <0.001 |
| **IFIS flexibility** | 282 | 2.8±0.9 | 1.9±0.8 | 2.7±0.7 | 3.3±0.8 | <0.001 |
| **Cardiovascular health score** | 265 | 51.9±12.8 | 43.8±9.6 | 50.3±13.3 | 56.3±11.5 | <0.001 |
| **Liver fat (%)[a]** | 247 | 7.4±11.9 | 15.7±18.3 | 9.2±11.7 | 5.2±8.2 | 0.001 |
| **Hs-CRP (mg/L)[a]** | 281 | 1.2±1.7 | 1.4±2.4 | 1.3±1.8 | 1.0±1.7 | 0.559 |
| **BMI (kg/m²)** | 273 | 29.4±4.5 | 32.7±4.4 | 30.2±4.5 | 27.5±3.6 | <0.001 |
| **Visceral adipose tissue volume (L)** | 261 | 6.1±2.6 | 7.6±2.7 | 6.5±2.5 | 5.2±2.5 | <0.001 |
| **Thigh fat-free muscle volume (Z-score)** | 258 | 0.02±1.05 | −0.04±1.12 | −0.10±1.01 | 0.16±1.06 | 0.155 |

Categorical variables are depicted as frequency (percentages). Continuous variables normally and not normally distributed (marked with [a]) are depicted as mean±standard deviation and median±interquartile range, respectively.

For descriptive comparison across IFIS overall fitness categories, P-values from one-way ANOVA (continuous variables) and chi-squared tests (categorical) were performed, without adjustment for multiple comparisons.

Incretin-based medications refer to the use of glucagon-like peptide-1 receptor agonists (GLP-1 RAs) and/or inhibitors of the enzyme dipeptidyl peptidase-4 (DPP-4 inhibitors).

Thigh fat-free muscle volume (Z-score) is standardized by sex, thigh length, height, and BMI from the general population [29].

BMI: body mass index, hs-CRP: high-sensitivity C-reactive protein, IFIS: International Fitness Scale, PA: physical activity (7-day wrist-worn accelerometry, ActiGraph® GT3X), min: minute, SGLT2: sodium-glucose co-transporter 2 medication, T2DM: type 2 diabetes mellitus.

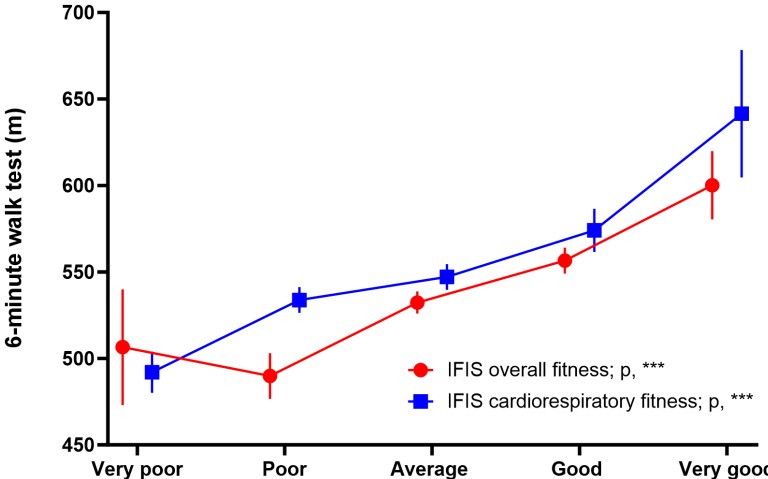

**Fig 2. Comparison between self-reported (IFIS) overall and cardiorespiratory fitness versus objective (6-minute walk) cardiorespiratory fitness.** The graphic depicts means with standard errors for the 6-minute walk test according to different levels of the IFIS scores. Analyses of covariance (ANCOVA) were adjusted for sex, and age. Significance level for each IFIS test is indicated as follows: ***, p<0.001; **, p<0.01; *, p<0.05; and ns, non-significant. IFIS: International Fitness Scale.

IFIS cardiorespiratory fitness and IFIS overall showed adequate criterion validity against the 6-minute walk test in participants without neuropathy/angiopathy (ANCOVA, both p < 0.05). However, among participants with neuropathy/angiopathy, neither IFIS cardiorespiratory fitness not IFIS overall showed adequate criterion validity, (ANCOVA, p = 0.171 and p = 0.117, respectively), S3 Fig and S4 Table.

### Self-reported physical fitness in relation to cardiometabolic and body composition outcomes

Fig 3 illustrates the associations between the IFIS scores (overall and cardiorespiratory fitness) and Z-scores of cardiometabolic (left panel) as well as body composition outcomes (right panel).

Both IFIS scores showed significant associations with Z-scores of cardiometabolic and, particularly, body composition outcomes (p<0.05 for ANCOVA in all comparisons). In general, better IFIS scores were associated with higher cardiovascular health scores and muscle volume, alongside lower liver fat content, high-sensitivity C-reactive protein, BMI, and visceral adipose volume. To provide context, we also examined associations of the 6-minute walk test with the same outcomes (see S4 Fig), which were broadly similar but not statistically significant for liver fat or high-sensitivity C-reactive protein (p>0.05). These analyses were not intended to position the 6-minute walk test as a body composition measure, but rather to contextualize the construct validity.

S5 Fig depicts the associations between additional IFIS scores (muscular fitness, speed-agility and flexibility) and Z-scores of cardiometabolic and body composition outcomes. While IFIS speed-agility and IFIS flexibility showed significant associations with all cardiometabolic and body composition outcomes (p<0.05 for ANCOVA in all comparisons), IFIS muscular fitness only showed significant associations with visceral adipose volume and muscle volume. S5 Table depicts the differences in cardiometabolic outcomes (Z-scores) according to categories of self-reported (IFIS) physical fitness scores.

### Predictive capacity of self-reported versus objective physical fitness for cardiometabolic and body composition outcomes

Fig 4 depicts the predictive performance for the IFIS scores (overall and cardiorespiratory) compared to the 6-minute walk test for cardiometabolic and body composition outcomes.

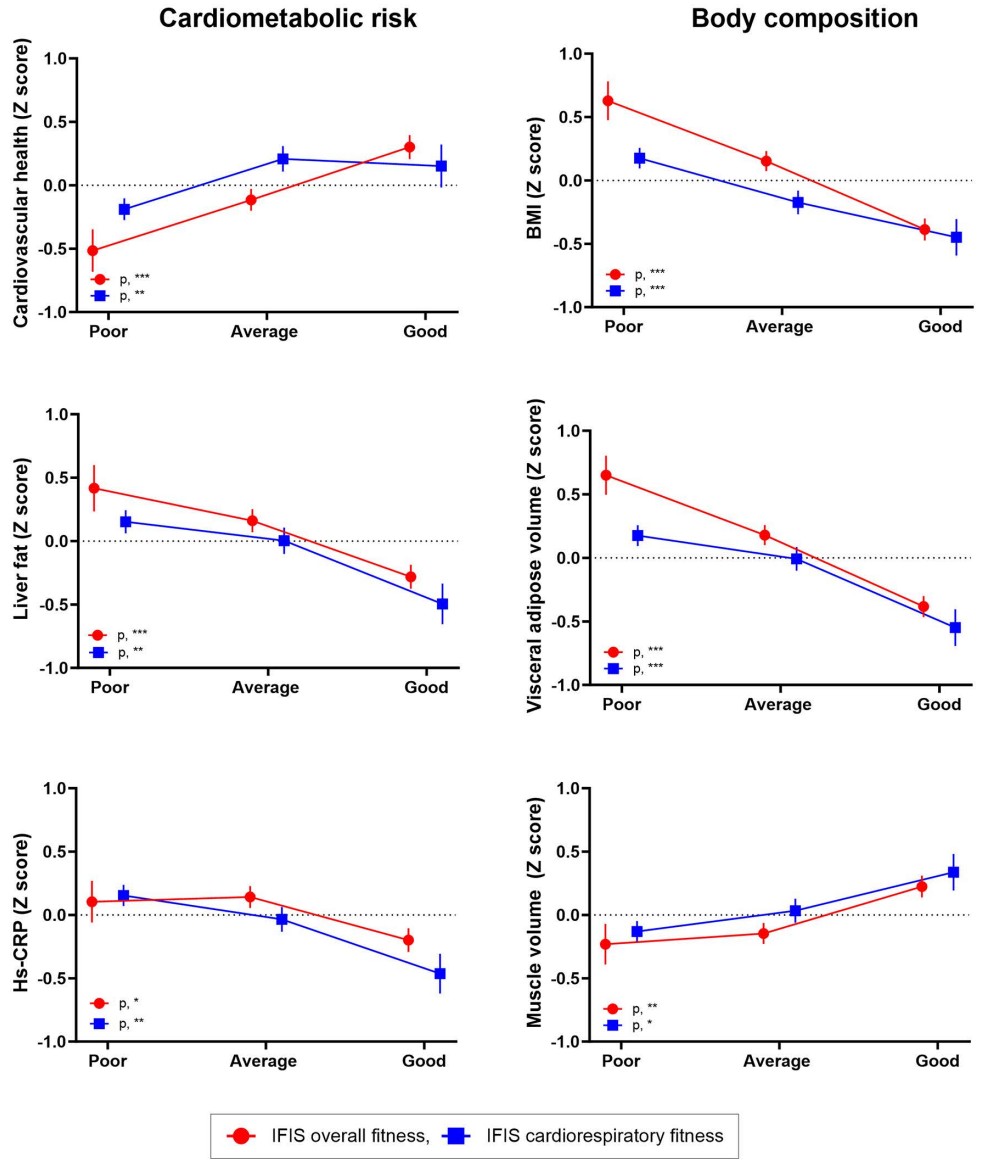

**Fig 3. Differences in cardiometabolic risk and body composition outcomes (Z-scores) according to categories of self-reported (IFIS) overall and cardiorespiratory fitness.** The graphic depicts means and standard errors for the analyses of covariance (ANCOVA) adjusted for sex, and age. The variables liver fat and hs-CRP were natural-logarithmically transformed. All variables are presented as study-specific Z-scores (mean = 0, standard deviation = 1). IFIS scores are categorized as poor (very poor/poor), average and good (good/very good). Significance level for each IFIS test is indicated as follows: ***, p < 0.001; **, p < 0.01; *, p < 0.05; and ns, non-significant. BMI: body mass index, hs-CRP: high-sensitivity C-reactive protein, IFIS: International Fitness Scale.

For cardiometabolic risk outcomes, the results showed that the predictive ability of the IFIS scores was at least as good as that of the 6-minute walk test. Notably, for predicting poor cardiovascular health, the IFIS overall fitness scores (AUC: 0.678, 95% CI: 0.618–0.737, p = 0.031) outperformed the 6-minute walk test (AUC: 0.586, 95% CI: 0.517–0.654). Similarly, the IFIS overall fitness scores (AUC: 0.630, 95% CI: 0.563–0.696, p = 0.016) outperformed the 6-minute walk test (AUC: 0.519, 95% CI: 0.443–0.595) for MASLD. For body composition outcomes, both IFIS scores demonstrated equivalent performance to the 6-minute walk test (p > 0.05 for these AUC comparisons).

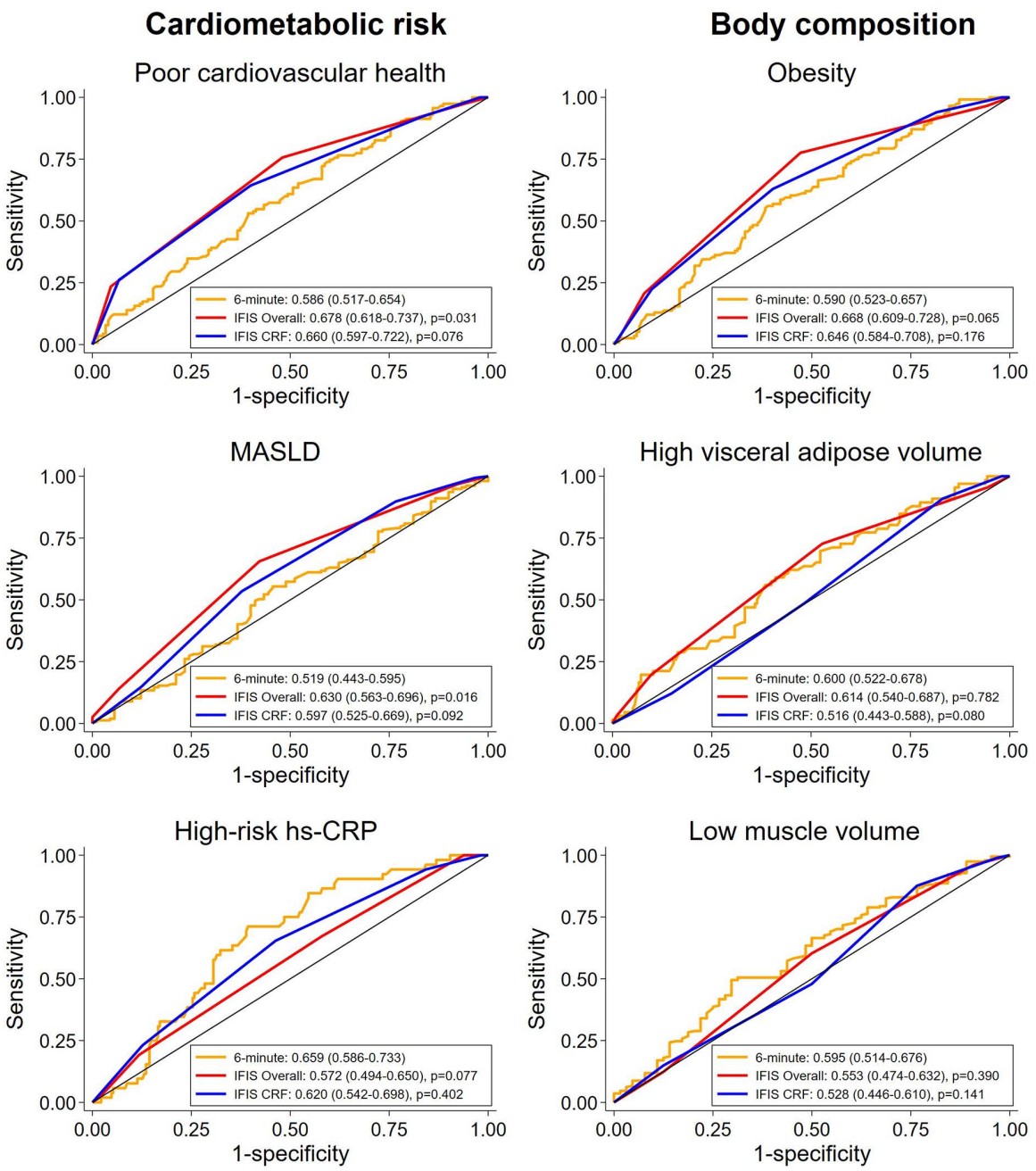

**Fig 4. ROC curves comparing self-reported (IFIS) overall and cardiorespiratory fitness versus objective (6-minute walk) cardiorespiratory fitness for predicting cardiometabolic risk and body composition outcomes.** The legends indicate the AUCs with corresponding 95% confidence intervals. The p-values represent comparisons of the AUCs between each specific IFIS test (IFIS overall and IFIS cardiorespiratory fitness) and the 6-minute walk test, performed using the DeLong test. Outcomes were dichotomized using established thresholds: poor cardiovascular health (<50 points), MASLD (≥5% liver fat), high-risk hs-CRP (>3.0 mg/L), obesity (BMI ≥ 30 kg/m²), high visceral adipose tissue (≥75th percentile), and low muscle volume (<25th percentile). AUC: area under curve, CRF: cardiorespiratory fitness, hs-CRP: high-sensitivity C-reactive protein, IFIS: International Fitness Scale, MASLD: metabolic dysfunction-associated steatotic liver disease, ROC: receiver operating characteristic curve.

Overall, a sensitivity analysis using alternative cut-offs to dichotomize the outcomes (according to the median of Z-scores instead of standardized clinical outcomes) rendered similar conclusions, although the predictive capacity of the IFIS cardiorespiratory score to predict muscle volume was lower to that of the 6-minute walk test (instead of being similar, as observed in the main analysis) (S6 Fig).

S7 Fig depicts the predictive performance of the IFIS scores probabilities compared to the 6-minute walk test probabilities after adjusting for age and sex. It shows that the predictive ability of the IFIS scores was similar to that for the 6-minute walk test in all cardiovascular risk and body composition outcomes. S6 Table depicts the differences in body composition outcomes (Z-scores) according to categories of self-reported (IFIS) physical fitness scores.

## Discussion

This study supports the usefulness of self-reported fitness, measured with the IFIS, as a proxy of objective physical fitness in adults with T2DM. First, both the IFIS overall and IFIS cardiorespiratory scores showed evidence of criterion validity, as they were positively associated with objectively measured performance on the 6-minute walk test. Additionally, IFIS overall and IFIS cardiorespiratory scores were both associated to all cardiometabolic outcomes (i.e., cardiovascular health, liver fat, and high-sensitivity C-reactive protein), and body composition outcomes (i.e., BMI, visceral adipose volume and muscle volume), showing evidence of construct validity. Furthermore, ROC analysis revealed that the IFIS scores performed as well as, or better than the 6-minute walk test in predicting poor clinical cardiometabolic and body composition outcomes. Specifically, the IFIS overall score outperformed the 6-minute walk test for predicting poor cardiovascular health and MASLD.

After its original validation in adolescents [10], the IFIS has demonstrated acceptable criterion validity against the 6-minute walk test in various adult populations, including women in early pregnancy [11], women with fibromyalgia [35], and older adults [36]. Our study expands this evidence to adults with T2DM. Consistent with previous findings, our study found that the IFIS cardiorespiratory score obtained a clear graded association with the 6-minute walk test. To contextualize, our study found that the average distance between consecutive categories on the IFIS cardiorespiratory fitness was approximately 32 meters, a difference that is clinically meaningful as it exceeds the threshold of approximately 30 meters considered relevant for middle-aged patients with T2DM [37].

Moreover, reporting less than "good" cardiorespiratory fitness (i.e., "very poor," "poor," or "average") on the IFIS yielded sensitivities of 93.9% and 87.5%, and negative predictive values of 82.1% and 94.4% for identifying individuals with reduced performance on the 6-minute walk test, defined as walking below the population-predicted mean distance or achieving less than 80% of their individualized predicted distance, respectively. Although the AUCs indicated poor discriminative ability (approximately 0.600), this may reflect the ordinal nature of IFIS, participants' tendency to avoid selecting the extreme categories ("very poor" or "very good"), and the considerable variation in 6-minute walk test distances among participants within the same IFIS category, rather than a true lack of association. The 6-minute walk test is also not a formal gold-standard test, which should be considered when interpreting these results. Nevertheless, the high sensitivity and negative predictive values suggest that IFIS may serve as a practical first-line screening tool to flag individuals for further assessment. To our knowledge, this is the first study to evaluate the discriminative performance of IFIS using ROC curves in adults with T2DM, providing clinically interpretable metrics for identifying individuals with reduced functional capacity. These findings support the criterion validity of the IFIS cardiorespiratory item while highlighting its limitations as a standalone diagnostic measure. Future studies incorporating ROC analyses in independent cohorts are warranted to refine the discriminative performance of IFIS, improve its generalizability, and explore strategies to enhance the resolution of self-reported fitness measures.

In addition to its criterion validity, several studies have established the construct validity of the IFIS, demonstrating associations with a range of health outcomes, including CVD risk factors (e.g., blood pressure, blood lipids, insulin resistance, inflammation, or metabolic syndrome) and adiposity indicators [38]. However, existing evidence on the IFIS largely

comes from studies on young and adolescent populations, with limited research exploring its validity in high CVD risk groups, particularly those with T2DM. By investigating the validity of the IFIS in middle-aged individuals with T2DM, who inherently face a heightened risk of CVD, this study may be of relevance for enhancing cardiovascular risk stratification.

In our study, IFIS cardiorespiratory fitness was associated with both body composition and cardiometabolic outcomes in a population with T2DM. Notably, IFIS overall fitness, a simple question regarding one's physical fitness relative to peers, also showed significant associations with all studied outcomes. Interestingly, our study also found that self-reported speed-agility and flexibility scores were clearly associated with cardiometabolic and body composition outcomes. In contrast, IFIS muscular fitness showed weaker associations, particularly with cardiometabolic outcomes and BMI. These findings aligns with previous literature across different populations [9,11], although such associations may vary depending on how muscular fitness is considered (i.e., absolute strength versus strength expressed relative to body weight) [9].

To the best of our knowledge, no previous study has directly compared the predictive ability of IFIS and objectively measured fitness for cardiometabolic and body composition outcomes. Importantly, our ROC analyses showed that self-reported overall and cardiorespiratory fitness scores predicted cardiometabolic and body composition outcomes at least as well as the 6-minute walk test. In fact, we observed that the IFIS overall fitness score outperformed the 6-minute walk test to predict poor cardiovascular health and MASLD, possibly due to IFIS overall fitness capturing multiple fitness dimensions, while the 6-minute walk test mainly reflects aerobic capacity. Though the capacity to discriminate poor cardiorespiratory performance was only moderate, the IFIS overall and cardiorespiratory fitness scores showed adequate construct validity to easily monitor not only physical fitness, but also cardiometabolic and body composition outcomes.

Cardiorespiratory fitness is supported by extensive epidemiological and clinical evidence as a strong predictor of mortality [39]. Additionally, incorporating cardiorespiratory fitness into risk assessments alongside traditional factors significantly enhances the precision of risk reclassification for adverse outcomes or mortality [39,40]. Recognizing the importance of this, the American Heart Association has identified cardiorespiratory fitness as a "vital clinical sign" and advocates for its routine assessment in clinical settings [13]. In this sense, while the IFIS showed limited ability to discriminate reduced functional capacity, it demonstrates acceptable criterion and construct validity, and remains a highly feasible option, requiring only a single self-reported item that can be completed in 1–2 minutes [10]. This is especially important in primary care settings managing patients with T2DM, where objective fitness assessments are often difficult to implement. In line with this, the current American Diabetes Association Standards of Care recommend evaluating physical activity and fitness as part of comprehensive lifestyle management, given their association with improved glycemic control, reduced cardiovascular risk, and enhanced treatment adherence [41,42].

A strength of the study lies in the use of high-quality methods, such as body composition outcomes analyzed using fat and water separated MRI and MRS for liver fat, ensuring the comprehensive assessment of cardiometabolic and body composition outcomes. Furthermore, the EPSOMIP cohort was drawn from unselected primary care patients with T2DM, including individuals across a wide range of ages, glycemic control levels, and comorbidity profiles. This enhances the external validity of our findings, making them generalizable to the broader population of adults with T2DM seen in routine clinical care.

Several limitations should be acknowledged. Although the cross-sectional design is appropriate for validation studies, enabling simultaneous assessment of the index test (IFIS) and reference standards, it inherently limits causal inference. Additionally, unmeasured confounders such as T2DM duration and treatment, diet, physical activity, socioeconomic status, and comorbidities may have influenced the observed associations. However, our objective was not to infer causality but rather to assess how well IFIS scores align with the 6-minute walk test and established cardiometabolic and body composition markers, supporting its construct and criterion validity in individuals with T2DM.

Test-retest reliability of IFIS was not assessed, though previous studies have demonstrated moderate to substantial reliability across populations [9,10,43], with minimal influence of learning or fatigue effects. Nonetheless, some heterogeneity in the reliability of the IFIS across populations must be acknowledged [44]. As a self-reported tool, the IFIS is subject

to response bias, potentially influenced by cognitive impairment, which could distort its association with objective health markers.

We did not compare IFIS scores to objective tests of muscular fitness, speed-agility, or flexibility, limiting the breadth of criterion comparisons. However, cardiorespiratory fitness, the component analyzed in greater depth in our study, remains the aspect of physical fitness most consistently associated with health outcomes [13].

Selection bias is a further consideration both from the initial participant recruitment and from missing data, as only 282 of 345 eligible participants had complete IFIS and 6-minute walk test results. Moreover, individuals with alcohol use disorder or high alcohol consumption were excluded based on screening questionnaires (2.3%), which may limit generalizability to this subgroup or reflect potential underreporting, given that this prevalence is unusually low for a Swedish population.

While the 6-minute walk test is a validated and practical field measure, it may not fully capture maximal cardiorespiratory capacity as laboratory-based assessments do, and may be influenced by varying motivation levels. Nonetheless, sensitivity analyses excluding low-effort performances yielded consistent results, supporting the robustness of our findings.

Finally, common T2DM-related complications such as neuropathy or angiopathy may also influence 6-minute walk test performance. Notably, while both IFIS cardiorespiratory and overall fitness demonstrated adequate criterion validity with the 6-minute walk test in participants without these conditions, neither IFIS cardiorespiratory fitness nor IFIS overall showed adequate criterion validity in those with angiopathy or neuropathy.

Further research is needed to assess the longitudinal validity of the IFIS and its predictive value for clinically relevant hard outcomes, including all-cause and CVD-related mortality and morbidity. Validation in independent T2DM cohorts is also necessary to confirm the generalizability of the ROC-based performance observed here. In addition, evaluating test–retest reliability and comparing IFIS with broader objective fitness measures would help determine its suitability for routine monitoring and clinical decision-making.

## Conclusions

Our findings extend previous research by demonstrating that the IFIS is a valid and feasible tool for assessing physical fitness in adults not only at low risk of CVD [9,10,38,45–48], but also at high risk, such as adults with T2DM. While the IFIS showed limited ability to discriminate reduced functional capacity, it nonetheless offers a practical tool for evaluating physical fitness in adults with T2DM. This is particularly relevant in settings such as primary care, where objective fitness assessments may be difficult to implement. Validated tools such as the IFIS could help identify patients with low fitness levels who may benefit from targeted physical activity interventions, potentially improving T2DM outcomes.

## Supporting information

**Appendix 1. Calculation of the Life's Essential 8 Factor score.**
(DOCX)

**S1 Fig. Distribution of responses to the five questions from the different self-reported (IFIS) scores.** IFIS: International Fitness Scale.
(TIF)

**S2 Fig. Sensitivity analysis comparing self-reported (IFIS) and objective (6-minute walk) physical fitness tests, including and excluding low-intensity 6-minute walk tests.** The graphic depicts means with standard errors for the 6-minute walk test according to different levels of the IFIS scores. Analyses of covariance (ANCOVA) are adjusted for sex and age. Presumably low-intensity 6-minute walk tests are considered those achieving less than 60% of the predicted maximal heart rate after test calculated with the Tanaka's formula. Significance level for each IFIS test is indicated as follows: ***, $p < 0.001$; **, $p < 0.01$; *, $p < 0.05$; and ns, non-significant. To allow comparability, population is restricted to those

having information on the heart rate after the 6-minute walk test, IFIS overall and IFIS CRF (n = 193), IFIS overall and IFIS CRF excluding low-intensity tests (n = 114). CRF: cardiorespiratory fitness, IFIS: International Fitness Scale.
(TIF)

**S3 Fig. Comparison of self-reported (IFIS) and objective (6-minute walk test) physical fitness between diabetic participants with and without neuropathy/angiopathy.** The graphic depicts means with standard errors for the 6-minute walk test according to different levels of the IFIS scores. Analyses of covariance (ANCOVA) are adjusted for sex and age. No neuropathy/angiopathy, n = 193. Neuropathy/angiopathy, n = 54. IFIS: International Fitness Scale.
(TIF)

**S4 Fig. Differences in cardiometabolic risk and body composition outcomes (Z-scores) according to percentiles groups of the 6-minute walk test.** The graphic depicts means and standard errors for the analyses of covariance (ANCOVA) adjusted for sex and age. The variables liver fat and hs-CRP were natural-logarithmically transformed. All variables are presented as Z-scores (mean = 0, standard deviation = 1). The 6-minute walk tests are categorized as <25th percentile, 25–75th percentile, and >75th percentile. Significance level for each 6-minute walk test is indicated as follows: ***, p < 0.001; **, p < 0.01; *, p < 0.05; and ns, non-significant. BMI: body mass index, hs-CRP: high-sensitive C-reactive protein, Pctl: percentile.
(TIF)

**S5 Fig. Differences in cardiometabolic risk and body composition outcomes (Z-scores) according to categories of self-reported (IFIS) physical fitness scores.** The graphic depicts means and standard errors for the analyses of covariance (ANCOVA) adjusted for sex and age. The variables liver fat and hs-CRP were natural-logarithmically transformed. All variables are presented as Z-scores (mean = 0, standard deviation = 1). IFIS scores are categorized as poor (very poor/poor), average and good (good/very good). Significance level for each IFIS test is indicated as follows: ***, p < 0.001; **, p < 0.01; *, p < 0.05; and ns, non-significant. BMI: body mass index, hs-CRP: high-sensitivity C-reactive protein, IFIS: International Fitness Scale.
(TIF)

**S6 Fig. Sensitivity analysis, ROC curves comparing self-reported (IFIS) overall and cardiorespiratory fitness versus objective (6-minute walk) cardiorespiratory fitness for predicting cardiometabolic risk and body composition outcomes (Z-scores), considering a different categorization of the outcomes.** The legends indicate the AUCs with corresponding 95% confidence intervals. The p-values represent comparisons of the AUCs between each specific IFIS test (IFIS overall and IFIS cardiorespiratory fitness) and the 6-minute walk test, performed using the DeLong test. Outcomes are dichotomized at the median of the Z-scores (0 indicating worse health, 1 indicating better health). AUC: area under curve, BMI: body mass index, CRF: cardiorespiratory fitness, hs-CRP: high-sensitive C-reactive protein, IFIS: International Fitness Scale, ROC: receiver operating characteristic curve.
(TIF)

**S7 Fig. ROC curves for the probabilities of adjusted logistic models based on self-reported (IFIS) overall and cardiorespiratory fitness versus objective (6-minute walk) cardiorespiratory fitness for predicting cardiometabolic risk and body composition outcomes.** The legends indicate AUCs with corresponding 95% confidence intervals. The p-values represent comparisons of the AUCs between the probabilities derived from the adjusted logistic models of the IFIS scores (IFIS overall and IFIS cardiorespiratory fitness) and the 6-minute walk test, performed using the DeLong test. All models are adjusted for age (continuous) and sex (female/male). Outcomes were dichotomized using established thresholds: poor cardiovascular health (<50 points), MASLD (≥5% liver fat), high-risk hs-CRP (>3.0 mg/L), obesity (BMI ≥ 30 kg/m²), high visceral adipose tissue (≥75th percentile), and low muscle volume (<25th percentile). AUC: area under curve, CRF: cardiorespiratory fitness, hs-CRP: high-sensitivity C-reactive protein, IFIS: International Fitness Scale, MASLD: metabolic dysfunction-associated steatotic liver disease, ROC: receiver operating characteristic curve.
(TIF)

**S1 Table. Clinical characteristics of the participants by IFIS overall fitness scores, 5 categories.** Categorical variables are depicted as frequency (percentages). Continuous variables normally and not normally distributed (marked with a) are depicted as mean ± standard deviation and median ± interquartile range, respectively. Incretin-based medications refer to the use of glucagon-like peptide-1 receptor agonists (GLP-1 RAs) and/or inhibitors of the enzyme dipeptidyl peptidase-4 (DPP-4 inhibitors). Thigh fat-free muscle volume (Z-score) is standardized by sex, thigh length, height, and BMI from the general population. BMI: body mass index, hs-CRP: high-sensitivity C-reactive protein, IFIS: International Fitness Scale, SGLT2: sodium-glucose co-transporter 2 medication, T2DM: type 2 diabetes mellitus.
(DOCX)

**S2 Table. Differences in the 6-minute walk test according to categories of self-reported (IFIS) physical fitness scores.** Data are presented as means and 95% confidence intervals. Adjusted models are adjusted by age and sex. Superscripts indicate statistically significant Tukey's pairwise comparisons (p < 0.05) for the means of the 6-minute walk test across categories of the IFIS scores: VP (Very poor), P (Poor), A (Average), G (Good), and VG (Very good). For example, in IFIS overall fitness, for the unadjusted model, those rating their overall fitness as "Poor" had significant differences in the 6-minute walk test compared to those rating their overall fitness as "Good" or "Very good". CI: confidence interval, IFIS: International Fitness Scale.
(DOCX)

**S3 Table. Diagnostic performance of the IFIS cardiorespiratory fitness test in identifying different levels of functional capacity as defined by the 6-minute walk test.** Population-predicted mean 6-MWT distance represents the weighted average estimated for the entire study population, calculated as 522.9 meters according to the equations by Enright & Sherrill, based on population-level age, weight, and height in women and men. Individualized predicted average 6-MWT distance is calculated for each participant using the Enright & Sherrill equations, based on their age, sex, weight, and height. 6-MWT: 6-minute walk test (meters), AUC: area under receiver operating characteristic curve, CI: confidence interval, DOR: diagnostic odds ratio, IFIS: International Fitness Scale, LR: likelihood ratio, NPV: negative predictive value, PPV: positive predictive value.
(DOCX)

**S4 Table. Differences in the 6-minute walk test according to categories of self-reported (IFIS) physical fitness scores comparing diabetic participants with and without neuropathy/angiopathy.** Data are presented as means and 95% confidence intervals. Adjusted models are adjusted by age and sex. Superscripts indicate statistically significant Tukey's pairwise comparisons (p < 0.05) for the means of the 6-minute walk test across categories of the IFIS scores: VP (Very poor), P (Poor), A (Average), G (Good), and VG (Very good). For example, in IFIS overall fitness, for the unadjusted model, those rating their overall fitness as "Poor" had significant differences in the 6-minute walk test compared to those rating their overall fitness as "Good" or "Very good". CI: confidence interval, IFIS: International Fitness Scale.
(DOCX)

**S5 Table. Differences in cardiometabolic outcomes (Z-scores) according to categories of self-reported (IFIS) physical fitness scores.** Data are presented as mean and 95% confidence intervals. Adjusted models are adjusted by age and sex. Superscripts indicate statistically significant Tukey's pairwise comparisons (p < 0.05) for the Z-scores of cardiometabolic outcomes across categories of the IFIS scores: P (Poor), A (Average), and G (Good). For example, in IFIS overall fitness, for the unadjusted model, those rating their overall fitness as "Poor" had significant differences in the Z-score for cardiovascular health compared to those rating their overall fitness as "Good". CI: confidence interval, hs-CRP: high-sensitivity C-reactive protein, IFIS: International Fitness Scale.
(DOCX)

**S6 Table. Differences in body composition outcomes (Z-scores) according to categories of self-reported (IFIS) physical fitness scores.** Data are presented as mean and 95% confidence intervals. Adjusted models are adjusted by age and sex. Superscripts indicate statistically significant Tukey's pairwise comparisons (p < 0.05) for the Z-scores of body composition outcomes across categories of the IFIS scores: P (Poor), A (Average), and G (Good). For example, in IFIS overall fitness, for the unadjusted model, those rating their overall fitness as "Poor" had significant differences in the Z-score for body mass index compared to those rating their overall fitness as "Average" or "Good". CI: confidence interval, IFIS: International Fitness Scale.
(DOCX)

## Acknowledgments

We gratefully acknowledge the EPSOMIP participants for their time and valuable contribution to the study.

We would like to thank Forum Östergötland for their support with study planning, legal guidance, and oversight of data quality. We also extend our gratitude to the coordinating study nurses, Carola Fagerström (Linköping) and Åsa Stahre Wiberg (Norrköping).

## Author contributions

**Conceptualization:** Ángel Herraiz-Adillo, Karin Rådholm, Carl-Johan Carlhäll, Olof Dahlqvist Leinhard, Peter Lundberg, Stergios Kechagias, Nils Dahlström, Patrik Nasr, Mattias Ekstedt, Pontus Henriksson.

**Data curation:** Ángel Herraiz-Adillo, Fredrik Iredahl, Peter Lundberg, Stergios Kechagias, Mattias Ekstedt.

**Formal analysis:** Ángel Herraiz-Adillo, Pontus Henriksson.

**Funding acquisition:** Carl-Johan Carlhäll, Olof Dahlqvist Leinhard, Peter Lundberg, Stergios Kechagias, Nils Dahlström, Patrik Nasr, Mattias Ekstedt, Pontus Henrikson.

**Investigation:** Ángel Herraiz-Adillo, Karin Rådholm, Fredrik Iredahl, Mikael Forsgren, Olof Dahlqvist Leinhard, Peter Lundberg, Nils Dahlström, Patrik Nasr, Mattias Ekstedt, Pontus Henriksson.

**Methodology:** Ángel Herraiz-Adillo, Karin Rådholm, Pontus Henriksson.

**Project administration:** Fredrik Iredahl, Peter Lundberg, Stergios Kechagias, Patrik Nasr, Mattias Ekstedt, Pontus Henriksson.

**Resources:** Mattias Ekstedt, Pontus Henriksson.

**Supervision:** Karin Rådholm, Carl-Johan Carlhäll, Olof Dahlqvist Leinhard, Peter Lundberg, Stergios Kechagias, Nils Dahlström, Patrik Nasr, Mattias Ekstedt, Pontus Henriksson.

**Validation:** Ángel Herraiz-Adillo, Karin Rådholm, Fredrik Iredahl, Carl-Johan Carlhäll, Nahuel Roemkens, Mikael Forsgren, Olof Dahlqvist Leinhard, Peter Lundberg, Stergios Kechagias, Nils Dahlström, Patrik Nasr, Mattias Ekstedt, Pontus Henriksson.

**Visualization:** Ángel Herraiz-Adillo, Pontus Henriksson.

**Writing – original draft:** Ángel Herraiz-Adillo, Pontus Henriksson.

**Writing – review & editing:** Ángel Herraiz-Adillo, Karin Rådholm, Fredrik Iredahl, Carl-Johan Carlhäll, Nahuel Roemkens, Mikael Forsgren, Olof Dahlqvist Leinhard, Peter Lundberg, Stergios Kechagias, Nils Dahlström, Patrik Nasr, Mattias Ekstedt, Pontus Henriksson.

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
