## [Decision Letter · Decision Letter 0]

8 Jul 2025

Dear Dr. Herráiz-Adillo,

Thank you for submitting your manuscript to PLOS ONE. After careful consideration, we feel that it has merit but does not fully meet PLOS ONE’s publication criteria as it currently stands. Therefore, we invite you to submit a revised version of the manuscript that addresses the points raised during the review process.

We look forward to receiving your revised manuscript.

Kind regards,

Neftali Eduardo Antonio-Villa, MD PhD

Academic Editor

PLOS ONE

Journal Requirements:

“The EPSOMIP study was supported by ALF Grants, Region Östergötland (ME, FI, PL, PN), unrestricted grants from GILEAD (ME) and Diapharma (SK), the Lion Research Grant from the Faculty of Medicine, Linköping University (PN), and the Swedish Research Council (VR 2020-04826 to PL). This specific study was additionally supported by ALF Grants, Region Östergötland (RÖ-999299 to PH).”

“ None declared.”

5. In the online submission form you indicate that your data is not available for proprietary reasons and have provided a contact point for accessing this data. Please note that your current contact point is a co-author on this manuscript. According to our Data Policy, the contact point must not be an author on the manuscript and must be an institutional contact, ideally not an individual. Please revise your data statement to a non-author institutional point of contact, such as a data access or ethics committee, and send this to us via return email. Please also include contact information for the third party organization, and please include the full citation of where the data can be found.

Additional Editor Comments (if provided):

The reviewers strongly suggest to make some clarifications to improve the content of the manuscript. Please, revise and submit a point-by-point review.

Reviewers' comments:

Reviewer's Responses to Questions

**Comments to the Author**

1. Is the manuscript technically sound, and do the data support the conclusions?

Reviewer #1: Partly

Reviewer #2: Yes

Reviewer #3: Partly

Reviewer #4: Yes

2. Has the statistical analysis been performed appropriately and rigorously?

Reviewer #1: I Don't Know

Reviewer #2: No

Reviewer #3: Yes

Reviewer #4: Yes

3. Have the authors made all data underlying the findings in their manuscript fully available?

Reviewer #1: Yes

Reviewer #2: Yes

Reviewer #3: No

Reviewer #4: No

4. Is the manuscript presented in an intelligible fashion and written in standard English?

Reviewer #1: Yes

Reviewer #2: No

Reviewer #3: Yes

Reviewer #4: Yes

Reviewer #1: Herraiz-Adillo and colleagues aim to compare the international fitness scale (IFIS) that rests on patient’s ratings of their performance in five domains to the 6-minute walk test that is a submaximal exercise test providing a distance in meters. Further, Herraiz-Adillo and colleagues compare IFS and 6-min walk test in terms of cardiometabolic and body composition outcomes as assessed by visceral adipose tissue volume, high sensitive C reactive protein, tight muscle volume and hepatic triglyceride concentration.

Herraiz-Adillo and colleagues conclude that IFIS is a valid tool for assessing physical fitness in diabetic patients and that it displays cardiometabolic and body compositions outcomes at least as well as the 6-minute walk test.

Major Comments:

Unless patients have serious cognitive impairment, one expects overall fitness and cardiorespiratory fitness assessments (from very poor, poor, average, good and very good) to somewhat correlate with walking distance over 6 minutes. (Fig 2)

While one expects the 6-minute walking distance to be somewhat indirectly related to body composition, the 6-minute walking test is not an appropriate tool to estimate body composition.

Minor Comments:

The study population is relatively small (282 subjects). Cardiorespiratory fitness depends on the daily activities of patients. Whether they were sedentary or physically active due to their occupations is not mentioned. Comparing physical performance ratings by an active security guard and an office worker to distances walked over 6 minutes may not be valid.

Patients with cardiac or pulmonary diseases are instructed to walk at a speed that they can sustain (submaximal test) and as much as possible to avoid any stops. The maximal heart rate is not an endpoint unless for comparison of multiple 6-minute walk tests.

Skeletal muscle intermuscular adipose tissue may have been a more sensitive endpoint that skeletal muscle volume (Miller Eur Heart J 2025)

In brief the data provided by Herraiz-Adillo and colleagues may be of interest to epidemiologists in need of fitness data and of lesser interest to physicians taking care of diabetic patients.

Reviewer #2: I am grateful for the opportunity to review this manuscript titled "Self-Reported Fitness (IFIS) in Adults with Type 2 Diabetes Mellitus: Validity and Associations with Cardiometabolic and Body Composition Outcomes." The study aims to evaluate the validity of the International Fitness Scale (IFIS) in assessing cardiorespiratory fitness compared to the 6-minute walk test and to explore their associations with cardiometabolic and body composition outcomes in adults with type 2 diabetes mellitus (T2DM), using a cross-sectional analysis of data from the EPSOMIP study.

Title:

Comment 1: (Page 1, Line 1) The title does not specify the study design as cross-sectional, which may mislead readers about the nature of the research, particularly since validity and associations could imply a longitudinal or experimental approach.

Comment 2: (Page 1, Line 1) The title mentions "validity" and "associations" but omits explicit reference to the comparison between self-reported fitness (IFIS) and an objective measure (6-minute walk test), which is a central focus of the study, thus lacking clarity on its primary intent.

Abstract:

Comment 3: (Page 1, Lines 27-28) The abstract states the aim "to assess the validity of the International Fitness Scale (IFIS) for evaluating cardiorespiratory fitness compared to the 6-minute walk test," but fails to specify that this refers to criterion validity, leaving an important methodological distinction unclear.

Comment 4: (Page 1, Lines 39-40) The results claim "strong correlations" between IFIS scores and the 6-minute walk test, yet no correlation coefficients or p-values are provided, rendering the strength of these associations vague and unsubstantiated within the abstract.

Comment 5: (Page 1, Line 46) The conclusion asserts that IFIS "conveys cardiometabolic and body composition outcomes at least as well as the 6-minute walk test," but this is not directly supported by the abstract’s results, which only discuss correlations and AUC comparisons, not a comprehensive equivalence in conveying outcomes.

Introduction

Comment 6: (Page 3, Line 52) The introduction notes the "rising prevalence" of T2DM but provides no specific statistics beyond a vague reference to "more than 500 million people" (Line 53), lacking precision and context to justify the global health concern.

Comment 7: (Page 3, Lines 53-54) The link between T2DM and lifestyle factors is mentioned, but there is no elaboration on how physical fitness specifically influences T2DM management or outcomes, weakening the rationale for focusing on fitness.

Comment 8: (Page 3, Lines 58-59) The claim that cardiorespiratory fitness reduces T2DM risk lacks a direct citation at this point (references [4,5] appear later), and its relevance to patients already diagnosed with T2DM is not established.

Comment 9: (Page 4, Lines 85-86) The objectives are stated, but the manuscript does not justify why cardiorespiratory fitness is prioritized over other fitness components (e.g., muscular fitness), despite IFIS assessing multiple domains.

Methods

Study Design:

Comment 10: (Page 4, Line 93) The cross-sectional design is mentioned, but there is no justification for its suitability to assess validity and associations, particularly given the inability to establish causality, which is a critical limitation for these objectives.

Comment 11: (Page 4, Lines 99-100) The recruitment period is provided, but the manuscript omits the total number of eligible participants and response rate, hindering assessment of selection bias and representativeness.

Participants:

Comment 12: (Page 5, Line 102) Inclusion criteria reference "current guidelines" for T2DM diagnosis, but these guidelines are not specified (e.g., ADA, WHO).

Comment 13: (Page 5, Lines 103-104) Exclusion criteria include "contraindication for performing Magnetic Resonance Imaging," but specific contraindications (e.g., pacemakers, claustrophobia) are not detailed.

Comment 14: (Page 5, Line 107) The sample size of 282 is reported, but the manuscript lacks a breakdown of exclusions by reason (beyond 18 withdrawals, Page 5, Line 106), obscuring the participant flow and potential biases.

Variables:

Comment 15: (Page 6, Lines 127-128) The IFIS is described, but its validation in Swedish or in adults with T2DM is not addressed, raising concerns about its cultural and clinical appropriateness for this population.

Comment 16: (Page 6, Lines 137-138) The 6-minute walk test is presented as a measure of cardiorespiratory fitness, but its limitations (e.g., influence of motivation, comorbidities like neuropathy) in T2DM patients are not discussed.

Data Sources & Measurement:

Comment 17: (Page 7, Lines 153-154) The cardiovascular health score (Life’s Essential 8) is introduced, but the manuscript lacks details on measurement methods for each component (e.g., blood pressure, lipids) and specific scoring cutoffs.

Comment 18: (Page 7, Lines 165-166) Liver fat measurement via PDFF is noted, but its accuracy or potential biases (e.g., in T2DM patients with altered liver metabolism) are not discussed, questioning its validity in this context.

Bias & Confounding:

Comment 19: (Page 8, Lines 196-197) Adjustments for sex and age are mentioned in ANCOVA models, but other potential confounders (e.g., T2DM duration, medication use, physical activity levels) are not addressed.

Comment 20: (Page 8, Lines 205-206) ROC curve use is described, but the manuscript does not consider the risk of overfitting or the need for external validation.

Sample Size & Statistical Methods:

Comment 21: (Page 5, Line 107) The sample size of 282 is stated, but no power calculation is provided to justify its adequacy for detecting differences or associations.

Comment 22: (Page 8, Lines 191-192) Statistical methods mention ANCOVA and ROC curves, but handling of missing data (e.g., for 247 participants with MASLD data vs. 282 total, Page 10, Line 234) is not explained.

Results

Comment 23: (Page 10, Line 227) Table 1 stratifies characteristics by IFIS overall fitness, but the manuscript lacks a full distribution of participants across all IFIS categories (e.g., all five levels).

Comment 24: (Page 10, Line 230) Mean age and other characteristics are reported, but differences across IFIS categories are not statistically tested or discussed.

Outcome Data & Main Results:

Comment 25: (Page 11, Lines 243-244) Fig 2 shows dose-response associations, but specific statistical measures (e.g., regression coefficients) are not provided, obscuring the magnitude and precision of these relationships.

Comment 26: (Page 12, Lines 269-270) "Significant associations" are reported, but p-values and confidence intervals are absent in the text (only in figures).

Comment 27: (Page 13, Lines 296-297) ROC analysis results include AUCs, but their clinical significance (e.g., practical utility of AUC 0.678 for cardiovascular health) or implications are not addressed.

Discussion

Comment 28: (Page 15, Lines 328-329) The claim that IFIS is a "valid tool" is based on cross-sectional data, which cannot fully establish validity (e.g., predictive or longitudinal validity).

Comment 29: (Page 15, Lines 337-338) IFIS outperforming the 6-minute walk test for some outcomes is noted, but reasons (e.g., self-report bias, test limitations) or implications are not explored.

Comment 30: (Page 17, Lines 396-397) The lack of test-retest reliability is acknowledged, but other biases (e.g., self-report bias in IFIS, selection bias from MRI exclusions) are not discussed.

Comment 31: (Page 17, Lines 404-405) The 6-minute walk test’s limitations as a proxy are mentioned, but its specific shortcomings in T2DM patients (e.g., neuropathy affecting walking) are not detailed.

Comment 32: (Page 16, Lines 385-386) Previous IFIS research is referenced, but specific comparisons to studies in similar populations (e.g., adults with chronic conditions) are absent.

Comment 33: (Page 16, Lines 390-391) The call for further studies is vague, lacking specific research questions or hypotheses (e.g., longitudinal validation, hard endpoints).

Reviewer #3: Review Plos One june 2025

The study evaluates the coherence between a cardiorespiratory fitness questionnaire and the 6 min walking test, while also assesses associations to cardiometabolically associated variables. The study is based on a sample from the Swedish primary care, and is interesting, while providing new insights on IFIS. However, the manuscript would benefit from being more clear in precisely what has been analysed, what was found, and what it potentially means, while avoiding bold statements.

Major:

1. Please clarify ”The American Heart Association recognizes cardiorespiratory fitness as a vital clinical sign”. How can a continous scale be considered a sign, especially of vitality? Are there established and valitated cutoffs which have clinical significance for specific tests and in what clinical settings are they used and for what purpose? This information should be considered also in the rest of the manuscript if the word ”clinical” is to be used. In general, the study would benefit from information on the validity of specific cutoffs, as this is often how tests are used in a clinical setting; ”good enough” or ”pathological”. If this questionnaire is currently used, how is it precisely used and to what benefit?

2. The authors argue that patients with ”conditions like peripheral neuropathy or peripheral arterial disease” may be unable to perform a physical assessment, which would indicate that this group may benefit from the FIS questionnaire instead. While these complications are common in T2D, it is very rare that they prevent a person from walking for 6 minutes. Furthermore, information on neuropathy and arterial disease in study participants is missing from this study. Therefore, this study does not validate the test for these individuals. Please rephrase the introduction as it is currently missleading. Also, please add the absence of this characterisation this to the limitation of the manuscript.

3. It is problematic to discuss that the FIS-questionnaire ”conveys” cardiovascular risk based on the current study design with cross-sectional associations to established risk factors for cardiovascular disease. It is likely better for any clinician to simply measure blood pressure, LDL and glucose instead of using the questionnaire. Please rephrase this to reflect that you find that the FIS-questionnaire ”associates” with risk factors to a similar magnitude as the walking test.

4. The study aims to ”validate” the IFIS questionnaire. However, it is not clearly stated what á priori defined criteria had to be met in order for the IFIS to be considered ”valid”. Please refer to general and established criteria for instrument validation which are typically used for validation of instruments in clinical use, and discuss your results according to effect sizes and approprietness of tests utilized in this study. Also, ”good criterion validity for assessing cardiorespiratory fitness when compared with objectively measured cardiorespiratory fitness tests, such as the 6-minute walk test” Please define what is ”good validity” is and how you concluded that the validity in this case was good.

5. If clinical validation is to be assessed, please consider to evaluate what sensitivity, specificity, PPV and NPV IFIS has to identify individuals with a poor performance of the 6 min walking test.

Minor:

1. The word ”proves” is used by the authors to state the valitidy of the quesitionnaire, which is a bit strong as a statement. While there is a correlation between the test and objective estimation of fitness in a selected group of patients with T2D in a single site observational study. That is not sufficient evidence to prove something. ”Indicates” may be more appropriate.

2. The authors argue that the FIS-questionnaire is easier to use than the 6-min walking test. Please clarify how long the FIS-test takes to complete to further support this claim.

3. Please communicate more clearly when it has meaning to objectively assess the cardiorespiratory fitness of a patient with T2D? Is this clinically used today or is this more of an instrument for academic purposes?

4. Patients with alcohol problems are excluded from the study. How was this defined? This may cause a problem to the external validity of the study, as 1/6 individuals in Sweden have a harmful use of alcohol.

5. How many were invited to participate in ESOMIP and what was the acceptance rate of the initial study?

6. Please clearly communicate if this was the primary endpoint of the ESOMIP-trial, a secondary endpoint, or an exploratory post hoc study. If this was a post hoc study, please also state that in the abstract.

7. Please explain the IFIS questionnaire in greater detail in methods. Time to complete, specifically trained staff or not, number of questions/points in total, established cutoffs or contious scale. You validate two estimates in the study, how are they derifed from the form?

8. ”Thus, the IFIS provides an accessible measure that may be easily integrated into clinical practice for physical fitness or cardiometabolic risk assessment in patients with an increased risk of CVD, such as individuals with T2DM.” Please exemplify when this would be meaningful in order to promote a direct effect on T2D treatment outcomes, based on current evidence and current T2D treatment guidelines from eg. ADA. If this is not possible, please abstain from using the terminology ”clinical practice”.

9. The contribution from the study participants should be acknowledged in acknowledgments.

10. The ROC curve AUC are generally low in this study (clinically relevant AUC are generally at least >95% in order to be clinically useful). This is OK as you only use them for relative comparisons between the IFIS and walking test. However, this should be clearly stated so that readers get a full understanding of the results.

11. It is difficult from the results to understand how you can use statements as ”good”, ”strong” and so on. If similar terms are to be used. Please use a methological reference which supports the statement that this particular effect size is generally considered good or strong. Otherwise, please soften up these statements.

12. Why are only 4 of the 8 variables in life’s essential eight used? I do not find support for this in the reference used. What is the purpose of selecting these parameters instead of the more diabetes-relevant measures HbA1c, blood pressure, LDL and BMI which are actually clinically used for all patients with T2D.

13. As you z-transform most variables, you examine how the variability of the variables associate with IFIS, while information on the absolute effect on the outcomes can not be evaluated. Please use a language which clearly communicates this.

Reviewer #4: Dear Dr. Herráiz-Adillo et al.,

Your study provides valuable evidence supporting IFIS as a practical tool for fitness assessment in T2DM. However, revisions are needed:

Expand Limitations: Discuss the impact of cross-sectional design, unmeasured confounders (e.g., diet, socioeconomic status), and the 6-minute walk test’s suitability as a comparator.

Reliability Data: Report test-retest reliability of IFIS in your cohort or explicitly state this as a study limitation.

Clinical Utility: Acknowledge that IFIS AUCs (0.63–0.68) indicate "moderate" discrimination—useful for screening but insufficient for definitive diagnosis.

Generalizability: Address the homogeneity of your cohort and its implications for global applicability.

Data Availability: Adhere to PLOS ONE policy by depositing data in a public repository or justifying restrictions ethically/legally.

After revisions, this work will be a significant contribution to diabetes management.

Sincerely,

**Do you want your identity to be public for this peer review?** For information about this choice, including consent withdrawal, please see our Privacy Policy

Reviewer #1: No

Reviewer #2: No

Reviewer #3: No

Reviewer #4: No

---

## [Author Response · Author response to Decision Letter 1]

11 Sep 2025

Dear Reviewers,

We are pleased to submit our revision to the original manuscript, “Self-Reported Fitness (IFIS) in Adults with Type 2 Diabetes Mellitus: Validity and Associations with Cardiometabolic and Body Composition Outcomes” PONE-D-25-22142.

We would like to sincerely thank the reviewers for their thoughtful and constructive feedback, which has greatly contributed to improving the clarity and overall quality of our work. As the manuscript is already quite comprehensive, we have aimed to incorporate the revisions as concisely as possible while ensuring that all key points are fully addressed. We hope that our responses and the corresponding changes meet the reviewers’ and the journal’s expectations.

All authors have approved the final version.

An itemized, point-by-point response to the reviewers’ comments is provided in the Response to Reviewers file. The line numbers refer to the track-changes version of the manuscript, in which modifications are highlighted in yellow and deleted paragraphs are marked in red.

Sincerely,

Linköping, Sweden, September 01, 2025

Ángel Herraiz-Adillo, Postdoctoral Researcher, Linköping University, Sweden

Pontus Henriksson, Senior Associate Professor, Linköping University, Sweden

---

## [Decision Letter · Decision Letter 1]

30 Nov 2025

and Its Associations With Cardiometabolic Health and Body Composition

Dear Dr. Herráiz-Adillo,

Thank you for submitting your manuscript to PLOS ONE. After careful consideration, we feel that it has merit but does not fully meet PLOS ONE’s publication criteria as it currently stands. Therefore, we invite you to submit a revised version of the manuscript that addresses the points raised during the review process.

We look forward to receiving your revised manuscript.

Kind regards,

Georgian Badicu, Ph.D

Academic Editor

PLOS ONE

Journal Requirements:

Additional Editor Comments:

The authors did not fully respond to one of the reviewers' comments regarding the article title:''The title does not specify the study design as cross-sectional, which may mislead readers about the nature of the research, particularly since validity and associations could imply a longitudinal or experimental approach''

Reviewers' comments:

Reviewer's Responses to Questions

**Comments to the Author**

Reviewer #3: All comments have been addressed

Reviewer #5: All comments have been addressed

Reviewer #6: All comments have been addressed

2. Is the manuscript technically sound, and do the data support the conclusions?

Reviewer #3: Yes

Reviewer #5: Yes

Reviewer #6: Yes

3. Has the statistical analysis been performed appropriately and rigorously?

Reviewer #3: Yes

Reviewer #5: I Don't Know

Reviewer #6: Yes

4. Have the authors made all data underlying the findings in their manuscript fully available?

Reviewer #3: No

Reviewer #5: Yes

Reviewer #6: Yes

5. Is the manuscript presented in an intelligible fashion and written in standard English?

Reviewer #3: Yes

Reviewer #5: Yes

Reviewer #6: Yes

Reviewer #3: All comments addressed to satisfaction. I have no additional comments. All my prints have good answers and revisions met my feedback

Reviewer #5: Dear Authors,

Thank you for the opportunity to review this manuscript. I am attaching some suggestions for improving the initial version of your investigation:

Abstract

Lines 31-60:

- The structure of the abstract appropriately highlights the objectives, methodology, and results of the research;

- The structure of the abstract highlights the issues of the topic addressed, also mentioning the main objective of the research.

Introduction

Therefore, the context and background of the research provides information about the field of study, the problems existing in it, and the progress made previously.

Lines 97-105 - In this part, it is recommended that, in addition to the objectives, the purpose of the study and the research hypotheses be specified.

Methods

In the section – methods – the purpose is described clearly, precisely and completely. Essential information is included.

Lines 134-139 - the study diagram illustrates the sequence of research stages and the relationships between them, providing a clear and structured image of the work process.

Discussion

- In this section, the results obtained in the study are contextualized, being detailed as theoretical and practical implications based on the relevant results of the present study. The limitations of the study and the strengths identified in the study are described in accordance with the relevance of the study.

Conclusions

Lines 525-541 – it is recommended to rewrite the conclusions because they are too extensive and some of them can be moved to the practical implications. To reformulate future research directions in correlation with the relevant results of the study.

Reviewer #6: All comments have been addressed properly, but consider and investigate belows vague ones:

reviewer 2 question: The title does not specify the study design as cross-sectional, which may mislead readers

about the nature of the research, particularly since validity and associations could imply a

longitudinal or experimental approach.

I think the authors do not respond this one correctly.

**Do you want your identity to be public for this peer review?** For information about this choice, including consent withdrawal, please see our Privacy Policy

Reviewer #3: No

Reviewer #5: No

Reviewer #6: No

---

## [Author Response · Author response to Decision Letter 2]

3 Dec 2025

Dear Reviewers,

We are pleased to submit our revision to the original manuscript, PONE-D-25-22142R2

“Validity of the International Fitness Scale (IFIS) and Its Associations With Cardiometabolic Health and Body Composition in Adults With Type 2 Diabetes: A Cross-Sectional Study”

We would like to sincerely thank the reviewers for their thoughtful and constructive feedback. We hope that our responses and the corresponding changes meet the reviewers’ and the journal’s expectations.

An itemized, point-by-point response to the reviewers’ comments is provided in the Response to Reviewers file. The line numbers refer to the track-changes version of the manuscript, in which modifications are highlighted in yellow and deleted paragraphs are marked in red.

Sincerely,

Linköping, Sweden, December 03, 2025

Ángel Herraiz-Adillo, Postdoctoral Researcher, Linköping University, Sweden

Pontus Henriksson, Senior Associate Professor, Linköping University, Sweden

Journal Requirements:

Authors:

We confirm that the reference list contains no retracted manuscripts and that no modifications to the references have been incorporated in the current revision. However, the reference list has been reordered to reflect the changes made in the manuscript.

Additional Editor Comments:

The authors did not fully respond to one of the reviewers' comments regarding the article title: ''The title does not specify the study design as cross-sectional, which may mislead readers about the nature of the research, particularly since validity and associations could imply a longitudinal or experimental approach''

Authors:

Thank you. In the previous revision, the article title was updated to ‘Validity of the International Fitness Scale (IFIS) and Its Associations With Cardiometabolic Health and Body Composition in Adults With Type 2 Diabetes: A Cross-Sectional Study’, which explicitly indicates the cross-sectional nature of the work. However, we would be happy to revise the title further if the editor considers it necessary.

Review Comments to the Author

Reviewer #3: All comments addressed to satisfaction. I have no additional comments. All my prints have good answers and revisions met my feedback

Authors:

Thank you for the constructive comments.

Reviewer #5:

Dear Authors,

Thank you for the opportunity to review this manuscript. I am attaching some suggestions for improving the initial version of your investigation:

Abstract

Lines 31-60:

- The structure of the abstract appropriately highlights the objectives, methodology, and results of the research;

- The structure of the abstract highlights the issues of the topic addressed, also mentioning the main objective of the research.

Authors:

Thank you for the appreciation of our work.

Introduction

Therefore, the context and background of the research provides information about the field of study, the problems existing in it, and the progress made previously.

Lines 97-105 - In this part, it is recommended that, in addition to the objectives, the purpose of the study and the research hypotheses be specified.

Authors:

Thank you for the comments. We have added the research hypotheses as suggested. Considering that the manuscript is already very long (> 5000 words) after revising the manuscript according to the comments in the first round of revisions, we suggest not adding a separate section on the study’s purpose, as it is already implicit in the aim and hypotheses. We remain open to further adjustments if requested by the editor.

Revised version:

In Introduction, lines 97-98:

“We hypothesized that the IFIS is a valid tool for assessing physical fitness in adults at high cardiovascular risk, particularly those with T2DM. Thus, the aims of this study were: 1) to investigate the validity of the IFIS self-reported physical fitness scores (i.e., overall and cardiorespiratory fitness) in assessing cardiorespiratory fitness considering the objectively measured 6-minute walk test as reference (criterion validity), and 2) to examine and compare the associations of the IFIS self-reported physical fitness scores and the 6-minute walk test with a wide range of cardiometabolic and body composition outcomes in adults with T2DM (construct validity).”

Methods

In the section – methods – the purpose is described clearly, precisely and completely. Essential information is included.

Lines 134-139 - the study diagram illustrates the sequence of research stages and the relationships between them, providing a clear and structured image of the work process.

Authors:

Thank you very much.

Discussion

- In this section, the results obtained in the study are contextualized, being detailed as theoretical and practical implications based on the relevant results of the present study. The limitations of the study and the strengths identified in the study are described in accordance with the relevance of the study.

Authors:

Thank you for your appreciation.

Conclusions

Lines 525-541 – it is recommended to rewrite the conclusions because they are too extensive and some of them can be moved to the practical implications. To reformulate future research directions in correlation with the relevant results of the study.

Authors:

Thank you for this helpful comment. As recommended, we have rewritten the Conclusions to make them more concise and focused. We have also reformulated the future research directions to align them more clearly with the results obtained in our analysis.

Revised version:

In Discussion, lines 479-487:

“In this sense, while the IFIS showed limited ability to discriminate reduced functional capacity, it demonstrates acceptable criterion and construct validity, and remains a highly feasible option, requiring only a single self-reported item that can be completed in 1–2 minutes.[10] This is especially important in primary care settings managing patients with T2DM, where objective fitness assessments are often difficult to implement. In line with this, the current American Diabetes Association Standards of Care recommend evaluating physical activity and fitness as part of comprehensive lifestyle management, given their association with improved glycemic control, reduced cardiovascular risk, and enhanced treatment adherence.[41,42]”

In Discussion, lines 528-533:

“Further research is needed to assess the longitudinal validity of the IFIS and its predictive value for clinically relevant hard outcomes, including all-cause and CVD-related mortality and morbidity. Validation in independent T2DM cohorts is also necessary to confirm the generalizability of the ROC-based performance observed here. In addition, evaluating test–retest reliability and comparing IFIS with broader objective fitness measures would help determine its suitability for routine monitoring and clinical decision-making.”

In Conclusions, lines 535-553:

“Our findings extend previous research by demonstrating that the IFIS is a valid and feasible tool for assessing physical fitness in adults not only at low risk of CVD,[9,10,38,45–48] but also at high risk, such as adults with T2DM. While the IFIS showed limited ability to discriminate reduced functional capacity, it nonetheless offers a practical tool for evaluating physical fitness in adults with T2DM. This is particularly relevant in settings such as primary care, where objective fitness assessments may be difficult to implement. Validated tools such as the IFIS could help identify patients with low fitness levels who may benefit from targeted physical activity interventions, potentially improving T2DM outcomes.”

Reviewer #6: All comments have been addressed properly, but consider and investigate belows vague ones:

Reviewer 2 question: The title does not specify the study design as cross-sectional, which may mislead readers about the nature of the research, particularly since validity and associations could imply longitudinal or experimental approach.

I think the authors do not respond this one correctly.

Authors:

Thank you. As we commented previously, we updated the article title to ‘Validity of the International Fitness Scale (IFIS) and Its Associations With Cardiometabolic Health and Body Composition in Adults With Type 2 Diabetes: A Cross-Sectional Study’, which explicitly indicates the cross-sectional nature of the work. However, we would be happy to revise the title further if the editor considers it necessary.

---

## [Editor Report · Decision Letter 2]

7 Dec 2025

Validity of the International Fitness Scale (IFIS)

and Its Associations With Cardiometabolic Health and Body Composition

in Adults With Type 2 Diabetes: A Cross-Sectional Study

PONE-D-25-22142R2

Dear Dr. Ángel Herráiz-Adillo,

We’re pleased to inform you that your manuscript has been judged scientifically suitable for publication and will be formally accepted for publication once it meets all outstanding technical requirements.

Kind regards,

Georgian Badicu, Ph.D

Academic Editor

PLOS One
---

## [Editor Report · Acceptance letter]

PONE-D-25-22142R2

PLOS One

Dear Dr. Herráiz-Adillo,

I'm pleased to inform you that your manuscript has been deemed suitable for publication in PLOS One. Congratulations! Your manuscript is now being handed over to our production team.

Kind regards,

on behalf of

Dr. Georgian Badicu

Academic Editor

PLOS One